# Studies of Conorfamide-Sr3 on Human Voltage-Gated Kv1 Potassium Channel Subtypes

**DOI:** 10.3390/md18080425

**Published:** 2020-08-13

**Authors:** Estuardo López-Vera, Luis Martínez-Hernández, Manuel B. Aguilar, Elisa Carrillo, Joanna Gajewiak

**Affiliations:** 1Laboratorio de Toxinología Marina, Unidad Académica de Ecología y Biodiversidad Acuática, Instituto de Ciencias del Mar y Limnología, Universidad Nacional Autónoma de México, Ciudad de México 04510, Mexico; 2Posgrado en Ciencias Biológicas, Facultad de Ciencias, Universidad Nacional Autónoma de México, Ciudad de México 04510, Mexico; lamh14@gmail.com; 3Laboratorio de Neurofarmacología Marina, Departamento de Neurobiología Celular y Molecular, Instituto de Neurobiología, Universidad Nacional Autónoma de México, Juriquilla, Qro. 76230, Mexico; maguilar@unam.mx; 4Programa de Becas Posdoctorales de la Dirección General de Asuntos del Personal Académico (DGAPA-UNAM), Universidad Nacional Autónoma de México, Ciudad de México 04510, Mexico; helittsa@gmail.com; 5Department of Biology, University of Utah, Salt Lake City, UT 84112, USA; jgajewiak@gmail.com

**Keywords:** *Conus spurius*, FMRFamide peptides, conorfamides, CNF-Sr3, *Shaker* K^+^ channels, Kv1 voltage-gated K^+^ channels

## Abstract

Recently, Conorfamide-Sr3 (CNF-Sr3) was isolated from the venom of *Conus spurius* and was demonstrated to have an inhibitory concentration-dependent effect on the *Shaker* K^+^ channel. The voltage-gated potassium channels play critical functions on cellular signaling, from the regeneration of action potentials in neurons to the regulation of insulin secretion in pancreatic cells, among others. In mammals, there are at least 40 genes encoding voltage-gated K^+^ channels and the process of expression of some of them may include alternative splicing. Given the enormous variety of these channels and the proven use of conotoxins as tools to distinguish different ligand- and voltage-gated ion channels, in this work, we explored the possible effect of CNF-Sr3 on four human voltage-gated K^+^ channel subtypes homologous to the *Shaker* channel. CNF-Sr3 showed a 10 times higher affinity for the Kv1.6 subtype with respect to Kv1.3 (IC_50_ = 2.7 and 24 μM, respectively) and no significant effect on Kv1.4 and Kv1.5 at 10 µM. Thus, CNF-Sr3 might become a novel molecular probe to study diverse aspects of human Kv1.3 and Kv1.6 channels.

## 1. Introduction

CNF-Sr3 is a peptide belonging to the Conorfamides family within the diverse classes of conotoxins [1]. Conotoxins are short peptide toxins, usually 25 amino acids in length, contained in the venom of the predator marine snails of the genus *Conus* [2]. A common characteristic of these peptides is the presence of cysteine residues in their primary structure that usually form disulfide bonds during their maturation. This feature has been used to name the toxins as conopeptides (disulfide-poor peptides: zero or one) or conotoxins (disulfide-rich peptides: two or more) [3,4,5]. CNF-Sr3 has no cysteine residues and its C-terminus is similar to the molluskan cardioexcitatory tetrapeptide Phe-Met-Arg-Phe-NH_2_, initially found in the class Bivalvia of the phylum Mollusca, and afterward in other invertebrate and vertebrate phyla [6]. CNF-Sr3 presents a high similitude in sequence with respect to CNF-Sr1 and CNF-Sr2; these conorfamides elicit hyperactivity upon injection in mouse [7,8]. CNF-Sr3 was tested on the complete *Drosophila* family of voltage-dependent K^+^ channels and it was shown to inhibit the *Shaker* subtype and not to have any significant effect on the other major families of voltage-activated K^+^ channels: *Shab*, *Shaw*, *Shal*, and *Eag* [1].

The *Shaker* K^+^ channel was identified by electrophysiological analyses of an unusual leg- shaking *Drosophila melanogaster* fruit fly strain (under ether anesthesia) [9]. Once a K^+^ channel gene was identified, additional voltage-activated potassium genes have been discovered in the fruit fly and in many other species, including humans [10,11,12,13]. In mammals, there are eight members related to the *Shaker* family (Kv1.1–Kv1.8). Mutations in some of these members lead to several diseases in humans. To mention just a few examples, changes in Kv1.1 channel may cause episodic ataxia type 1 [14], whereas the Kv1.3 channel has been associated with multiple sclerosis, type-1 diabetes, and rheumatoid arthritis, but relatively little is known about Kv1.2, Kv1.4, Kv1.5, Kv1.6, Kv1.7, or Kv1.8 channels in human disorders [15]. However, Martel et al., by electrochemical recordings at rat dorsal striatum, confirmed a role for Kv1.2 and Kv1.6 in dopamine release [16] and other authors have shown that Kv1.4 and Kv1.5 are related to cardiac effects [17,18,19].

The present study arises from our previous finding that the *Shaker* K^+^ channel is inhibited by CNF-Sr3. Here, we report the effects of CNF-Sr3 on human Kv1.3, Kv1.4, Kv1.5, and Kv1.6 channels.

## 2. Results

### Evaluation of CNF-Sr3 on Kv1 channels Expressed in Oocytes

Previously, it was reported that CNF-Sr3 reversibly inhibits the *Shaker* K^+^ channel with no activity over other *Drosophila* families of voltage-dependent K^+^ channels: *Shab*, *Shaw*, *Shal*, and the distant h*Eag* K^+^ channel [1]. Therefore, we decided to examine the effect of CNF-Sr3 on four members of the human *Shaker*-related subfamily, the Kv1.3, Kv1.4, Kv1.5, and Kv1.6 channels, due to their functional implications in human diseases, particularly for Kv1.3 [20]. Since CNF-Sr3 has a Kd = 2.7 µM for the *Shaker* channel and there is evidence that some conotoxins, like κ−PVIIA, bind to *Shaker* but not to Kv1 channels [18,19], we decided to test CNF-Sr3 at higher final concentrations from 10 to 30 µM. At 10 µM, CNF-Sr3 was capable of abolishing up to 20% and 80% of the current for Kv1. 3 and Kv1. 6, respectively (Figure 1A,D), but decreased the currents for Kv1.4 and Kv1.5 channels by less than 5% (Figure 1B,C). Therefore, we decided to increase the concentration in those which were most sensitive (Kv1.3 and Kv1.6) and we found that almost all the current was inhibited irreversibly at 30 µM. In Figure 1E,F is shown the time course of inhibition of Kv1.3 and Kv1.6 by CNF-Sr3, respectively. Then, concentration–response curves were generated for the Kv1.3 and Kv1.6 channels. CNF-Sr3 blocked Kv1.6 with an ~10-fold greater affinity than Kv1.3 (IC_50_ 2.7 ± 0.89 µM and 24 ± 4.06 µM, respectively; mean ± SEM; n=3) (Figure 1G) and the Hill equation yielded n=1.08 for Kv1.6 and n=2.1 for Kv1.3. 

## 3. Discussion

In this work, we provide more evidence on the effect of CNF-Sr3 on voltage-gated potassium channels; specifically, we studied the effect on the human Kv1.3–Kv1.6 subtypes, closely related to the *Shaker* channel. We decided to start studying the effect of CNF-Sr3 on these subtypes because of their similarity to the *Shaker* channel, their impact on several diseases in humans, and because few conotoxins target these subtypes preferentially [20,21]. Until now, 21 conotoxins or conopeptides (including CNF-Sr3) have been reported to have activity on K^+^ channels [21,22,23,24]; 17 of them are disulfide-containing peptides, but four, including CNF-Sr3, do not contain Cys residues; interestingly, all of them are expressed by vermivorous species (Mo1659, by *C. monile* [25]; CPY-Fe1, by *C. ferrugineus* [26]; CPY-Pl1, by *C. planorbis* [26]; and CNF-Sr3, by *C. spurius* [1]). The first and most studied conotoxin with activity on K^+^ channels is κ-PVIIA isolated from the venom of *C*. *purpurascens*, which belongs to the superfamily O1. κ-PVIIA has an IC_50_ of 60 nM for the *Shaker* K^+^ channel expressed in oocytes, with no activity for the mammalian Kv1.1 and Kv1.4 subtypes [27]. Additionally, superfamilies A, M, J, and I_2_ have toxins that block the *Shaker* channel in the μM range, with a little effect on Kv1 channels [20,25,26,27,28,29,30]. In this sense, only κM-RIIIJ and κJ-PlXIVA, isolated from *C*. *radiatus* and *C*. *planorbis* (fish- and worm-hunting snails, respectively) have inhibitory activity on Kv1.6 channels with IC_50_ of 8 µM and 1.59 µM, respectively, whereas κ-ViTx, isolated from *C*. *virgo* (worm-hunting snail), has an IC_50_ of 2.09 μM for the Kv1.3 channel. ViTx has 31% amino acid percent identity with SrXIA, the very first conotoxin isolated from *C*. *spurius* with activity on Kv1 channels. SrXIA blocks 58% of the current elicited by Kv1.6 at 640 nM after 23 min in the recording chamber [31].

Like CNF-Sr3, conopeptides CPY-Fe1 and CPY-Pl1, isolated form *C*. *ferrugineus* and *C*. *planorbis*, respectively, also block the Kv1.6 subtype, with the latter having a higher affinity (IC_50_ = 170 nM), compared to CPY-Fe1 (IC_50_ = 8.8 µM) [26].

In this work, we show that CNF-Sr3 is capable of inhibiting the human Kv1.6 and Kv1.3 channels at the same order of magnitude, like conotoxins or conopeptides previously reported to have activity on voltage-dependent K^+^ channels. Further characterization will be needed to understand the mechanism of action of this toxin, such as determination of I/V curves to assess whether the binding of CNF-Sr3 is voltage-dependent.

A possible explanation for the low affinity of this toxin against Kv1 channels in this work might be the use of mammalian K^+^ channels. The effect of CNF-Sr3 may be expected to be in the μM range, as it was demonstrated for κM-RIIIK, whose IC_50_ for the *Shaker* channel is 1.21 μM and this value decreases one order of magnitude when determined on TSha1 from the fish *O*. *mykiss* (IC_50_ = 70 nM) [32]. 

Even though CNF-Sr3 has a rather low affinity for human Kv1 channels, its short sequence (constituted by 15 amino acid residues, with no cysteine residues), in comparison with the conotoxins or conopeptides described above, which have an average of 25 amino acids in length with four to eight cysteine residues, might allow it to become a useful molecular tool to elucidate the role of Kv1.6 and Kv1.3 channels in human and other mammalian species, and/or as a scaffold for the design and synthesis of more specific probe molecules for these channels. Regarding conotoxins without Cys residues, CNF-Sr3 might have an advantage as a molecular tool over CPY-Pl1 and CPY-Fe1, because it does not induce a leak current in skinned *Xenopus* oocytes (and, probably, other cell types) [26].

## 4. Materials and Methods 

### 4.1. Synthetic CNF-Sr3

CNF-Sr3 was synthesized using an Apex 396 automated peptide synthesizer (AAPPTec; Louisville, KY, USA) applying standard solid-phase Fmoc (9-fluorenylmethyloxy-carbonyl) protocols and quantified as was described recently [1].

### 4.2. Electrophysiology Assay in Oocytes 

cDNA of the clones for human Kv1.3 to Kv1.6 channels in vector pSGEM, resistant to ampicillin, were linearized with NotI and SpHI. Then, the products were purified using a QIAquick PCR purification Kit according the protocol by Qiagen (Hilden, Germany). cDNAs were transcribed in vitro with the T7 polymerase (mMessage mMachine Kit; Ambion Inc., Austin, TX, USA). cRNA was purified using a Qiagen RNeasy kit (Hilden, Germany). Five nanograms of cRNA was injected into each oocyte and used for voltage-clamp recording 1–2 days after injection. Sexually mature *Xenopus leavis* female frogs were kept and handled according to the institutional bioethics committee requirements. Whole-cell currents were recorded under two-electrode voltage clamp control using an OC-725C clamp amplifier (Warner Instruments, CT, USA). The intracellular electrodes were filled with 3 M KCl. Potassium currents were generated by de-polarizations to 10 mV for 1 s every 20 s with a membrane potential held at -80 mV. The bath solution was ND96, consisting of 96 mM NaCl, 2.0 mM KCl, 1.8 mM CaCl_2_, 1.0 mM MgCl_2_, and 5 mM HEPES (pH 7.5). CNF-Sr3 was diluted in ND96 and directly applied (3 μL) to the (static) bath chamber (30 μL). All electrophysiological measurements were performed at room temperature (~22 °C).

Measurements are reported as the mean ± SEM of three independent experiments. Dose–response curve data were fit to the equation: IC_50_ = fraction of channels blocked (fb) as a function of [CNF-Sr3], at 0 mV. fb = 1 − (I_CNF-Sr3_/IC), where I_CNF-Sr3_ toxin is the current in the presence of CNF-Sr3 in log μM. IC is the control IK current (SigmaPlot, Systat Software Inc., San Jose, CA, USA). 

## Figures and Tables

**Figure 1 marinedrugs-18-00425-f001:**
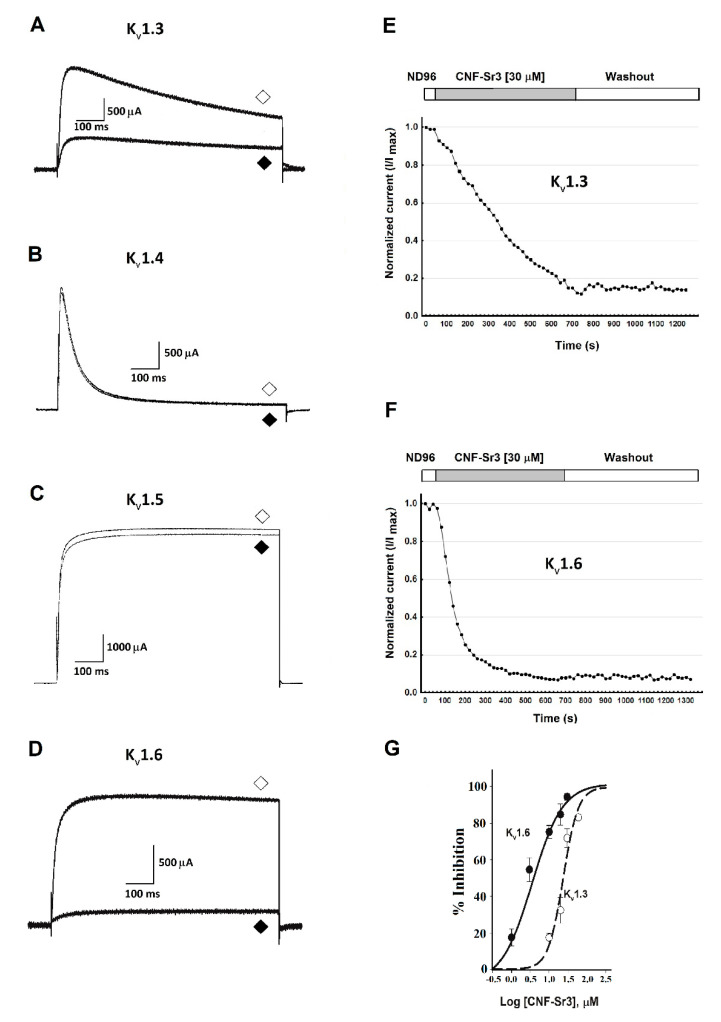
Inhibition of human K_v_1 channels by Conorfamide-Sr3 (CNF-Sr3). (**A**) K_v_1.3, control current (◇); current in presence of CNF-Sr3 **30 µM** (◆). (**B**) K_v_1.4, control current (◇); current in presence of CNF-Sr3 **10 µM** (◆). (**C**) K_v_1.5, control current (◇); current in presence of CNF- Sr3 **10 µM** (◆). (**D**) K_v_1.6, control current (◇); current in presence of CNF- Sr3 **30 µM** (◆). (**E**,**F**) Time course of inhibition by CNF-Sr3 on Kv1.3 and Kv1.6, respectively. CNF-Sr3 takes twice as much to block the current for Kv1.3 than Kv1.6. (**G**) CNF-Sr3 only inhibited Kv1.6 and Kv1.3 channels with low affinity: IC_50_ of 2.7 ± 0.89 µM and 24 ± 4.06 µM, respectively (n=3 for each concentration; error bars = standard error of the mean (SEM).

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
