# Peer review of "Studies of Conorfamide-Sr3 on Human Voltage-Gated Kv1 Potassium Channel Subtypes"

_marinedrugs, 2020, doi:10.3390/md18080425_

Round 1

Reviewer 1 Report

This manuscript appears to be an addendum to a previously publication by the authors on conorfamide-Sr3 (CNF-Sr3) inhibition of Shaker K+ channel (Campos-Lira et al., 2017, Toxicon, 139). In the present study, the activity of CNF-Sr3 was tested on four human voltage-gated K channel subtypes and shown to inhibit only Kv1.6 and Kv1.3 subtypes. The work appears to neatly carried out but the amount of new information is minimal. Unless there is something remarkable about the findings, two low affinity concentration-response curves for Kv1.6 and Kv1.3 may not be sufficient. Further details of the kinetics, voltage-dependence and reversibility of inhibition by CNF-Sr3 should be given. The effect of CNF-Sr3 on the four Kv1 channel subtypes is only shown for a single voltage and a current-voltage relationship for Kv1.6 and Kv1.3 in the absence and presence of the CNF-Sr3 should be shown. The authors should also show the time course of onset and recovery from block by CNF-Sr3 for completeness.

Author Response

Dear Reviewer

Thanks for your comment and we agree that the amount of new information is not the ideal. For this reason, we submitted the results as a short communications instead of Original research. However, we consider that the relevance of our work lies in the finding of inhibition selectivity for one of four subtypes of the hKv1 family. In addition, the fact that the CNF-Sr3 is a non-cysteines containing conopeptide of only 15 aa in length suggests that it could be useful as scaffold for the design of specific inhibitor molecules for these channels.

I/V curves were not determined because our previous results on the Shaker channel showed that the effect of CNF-Sr3 is not voltage-dependent (Campos-Lira et al., “Conorfamide-Sr3, a structurally novel specific inhibitor of the Shaker K+ channel,” Toxicon, vol. 138, 2017, doi: 10.1016/j.toxicon.2017.07.024).

Reviewer 2 Report

The manuscript of Lopez-Vera et al. explores the effect of the toxin CNF-Sr3 on human orthologues of the Shaker-type Kv1 family. From the selected members tested the data shows that hKv1.6 displays the highest affinity, although in the μM range. The report of a hKv1.6 specific toxin would be valuable but this cannot be concluded as not all Kv1 channels have been evaluated.

Major comment

1) The authors evaluated just 4 members of the large Kv1 family, which is composed of 8 members in total. The conclusion that the toxin CNF-Sr3 can be used to study hKv1.6 specifically is only valid if all members are evaluated, especially also hKv1.1 and hKv1.2 as several toxins (for example dendrotoxin, several scorpion toxins) have been reported to selectively inhibit hKv1.6, hKv1.1 and hKv1.2. So all members should be evaluated and hKv1.6 should display the highest sensitivity.

2) The current recordings of hKv1.5 in the figure indicate that 10 μM induces a small effect, what happens if higher concentrations are evaluated?

Minor comments

1) The data of the current recordings in the figure are misleading because for the four channels a different concentration is shown. For both hKv1.3 and hKv1.6 a 30 μM concentration was used whereas for hKv1.5 and hKv1.4 it is only 10 μM.

2) The statement that CNF-Sr3 has no effect on hKv1.4 and hKv1.5 is not fully correct, there was no or minimal effect with a CNF-Sr3 concentration up to 10 μM.

3) The background information on the Kv1 channels is short and for example hKv1.5 has been reported to be associated with atrial fibrillation. If the results show that CNF-Sr3 is selective for hKv1.6 compared to the other Kv1 channels, the background information can be more focused on hKv1.6.

4) Layout of the figure can be changed as it takes excessive space in its current form. The example recordings can be represented smaller.  

5) Panel e in the figure shows the concentration-effect curve whereby the effect is fraction inhibition which should be in % ranging from 0 to 100. Since the actual concentration is known, it is preferred to talk about concentration instead of a dose.

6) For the concentration effect curves only the IC50 values are reported but what were the slope factors (Hill number)?

7) The title is not very informative, “further study of” can mean a lot.

8) If the oocytes were surgically obtained from Xenopus Laevis frogs, check if approval was needed for the surgery protocol by the institutional ethics committee for animal experiments.

Author Response

Dear reviewer.

We would have liked to test CNF-Sr3 in all the members of hKv1 family. However, we unfortunately do not have all the clones to be expressed in Xenopus oocytes. Nevertheless, our findings might provide an initial guideline to develop new scaffolds for designing specific molecules for these channels. The fact that CNF-Sr3 is just 15 aa in length and does not contain cysteine residues, compared with dendrotoxins which include 57-60 aa residues crosslinked by three disulfide bridges, could be an advantage for the synthesis of specific inhibitory peptide molecules for the members of hKv1 family.

2) The current recordings of hKv1.5 in the figure indicate that 10 μM induces a small effect, what happens if higher concentrations are evaluated?

We think that higher concentrations might cause greater effects on hKv1.5, but the small effect shown by CNF-Sr3 at 10 µM was irrelevant according to our experience. For this reason, we did not pursue to test higher concentrations and because we wanted focus on the clear inhibition on hKv1.6.

Minor comments

1) The data of the current recordings in the figure are misleading because for the four channels a different concentration is shown. For both hKv1.3 and hKv1.6 a 30 μM concentration was used whereas for hKv1.5 and hKv1.4 it is only 10 μM.

We apologize for the confusion, even when we specified the concentration differences in the result section and in the figure caption. To avoid this problem, we have used boldface for the concentration values in the figure caption.

2) The statement that CNF-Sr3 has no effect on hKv1.4 and hKv1.5 is not fully correct, there was no or minimal effect with a CNF-Sr3 concentration up to 10 μM.

We apologize for this mistake; we have changed the sentence to “…no significant effect on Kv1.4 and Kv1.5 at 10 µM.” (in the Abstract).

3) The background information on the Kv1 channels is short and for example hKv1.5 has been reported to be associated with atrial fibrillation. If the results show that CNF-Sr3 is selective for hKv1.6 compared to the other Kv1 channels, the background information can be more focused on hKv1.6i

We appreciate the information about the implications of Kv1.5 on atrial fibrillation; we included a reference in the manuscript (Introduction section) about it. However, there is little or specific information for Kv1.6 in human disorders. For that reason, we do not abound on the matter.

4) Layout of the figure can be changed as it takes excessive space in its current form. The example recordings can be represented smaller.

We appreciate your comment again and it has already been modified.

5) Panel e in the figure shows the concentration-effect curve whereby the effect is fraction inhibition which should be in % ranging from 0 to 100. Since the actual concentration is known, it is preferred to talk about concentration instead of a dose.

We appreciate your recommendation and we changed the Y axis in % ranging from 0 to 100.

6) For the concentration effect curves only the IC50 values are reported but what were the slope factors (Hill number)?

We incorporated de Hill numbers for each curve at the end of the 2.1 Results section.

7) The title is not very informative, “further study of” can mean a lot.

We agree with you and we changed the title to “Studies of Conorfamide-Sr3 on human voltage-gated Kv1 potassium channel subtypes.”

8) If the oocytes were surgically obtained from Xenopus Laevis frogs, check if approval was needed for the surgery protocol by the institutional ethics committee for animal experiments.

We thank this important issue and we have the approval by the institutional ethics committee for animal experiments.

Reviewer 3 Report

In this manuscript, López-Vera et al. showed that CNF-Sr3 isolated from see snail venom suppressed voltage-gated K+ channel subtypes Kv1.3 and Kv1.6 but not Kv1.4 and Kv1.5. Many K+ channel toxins are sufficient to block the channels with submicromolar concentrations. However, the potency of CNF-Sr3 was not so attractive. There are several concerns to need to be addressed. I strongly recommend the careful reading by all authors.

  1. Authors should change the title. ‘Studies of conoefamide-Sr3 on human voltage-gated Kv1 potassium channel subtypes’
  2. In ‘Results’, authors should describe whether inhibitory effects of CNF-Sr3 on Kv1.3 and Kv1.6 are recovered by washout.
  3. In ‘Methods’, authors described that depolarization duration was 20 sec. But, in Figure 1, duration is about 1 sec.
  4. In Figure 1, current amplitude is huge. Usually, current amplitude in Xenopus oocyte expression system is under several microA. In this work, the current amplitudes at +10 mV were mA order.
  5. Page 5, line 144: Remove ‘at least’. In Figure 1, authors described ‘n=3’.

Author Response

Dear Reviewer

1. Authors should change the title. ‘Studies of conoefamide-Sr3 on human voltage-gated Kv1 potassium channel subtypes’

We appreciate the recommendation and we have changed the title according to it.

2. In ‘Results’, authors should describe whether inhibitory effects of CNF-Sr3 on Kv1.3 and Kv1.6 are recovered by washout.

We have incorporated a time course of inhibition by CNF-Sr3 for both Kv1.6 and Kv1.3 channels in Figure 1E and 1F. In section 2.1 we described the irreversibility of the effect.

3. In ‘Methods’, authors described that depolarization duration was 20 sec. But, in Figure 1, duration is about 1 sec.

We apologize for the mistake; we skipped to write down how often the de-polarizations were applied. Thus, we changed the corresponding sentence to “… by de-polarizations to 10 mV for 1 s every 20 s”

4. In Figure 1, current amplitude is huge. Usually, current amplitude in Xenopus oocyte expression system is under several microA. In this work, the current amplitudes at +10 mV were mA order.

The current amplitudes depend on diverse differences among oocyte batches; sometimes we have very small currents or even huge currents at +10 mV, that difficult the recordings. So, we have standardized for Kv1 channels to place the electrophysiology recordings 24 h upon injections of the RNAm encondig the Kv1 channels to have the major and similar current amplitudes for testing toxins or other compounds. Even the currents are in mA the current amplitudes are consistently steady among distinct oocytes.

5. Page 5, line 144: Remove ‘at least’. In Figure 1, authors described ‘n=3’.

We appreciate the remark about it, and ‘at least’ was deleted.

Round 2

Reviewer 1 Report

The revised manuscript is improved with the additional information and data provided with regard kinetics and lack of reversibility. Confirmation of I-V curves carried out on human Kv1.3 and Kv1.6 channels would be preferable rather than relying on the previous study of the Shaker K+ channel showing that the effect of CNF-Sr3 was not voltage-dependent. If the inhibition of Kv1.3 and Kv1.6 by CNF-Sr3 was not voltage-dependent, then this should be shown or stated in the Results.

Author Response

Comments and Suggestions for Authors

The revised manuscript is improved with the additional information and data provided with regard kinetics and lack of reversibility. Confirmation of I-V curves carried out on human Kv1.3 and Kv1.6 channels would be preferable rather than relying on the previous study of the Shaker K+channel showing that the effect of CNF-Sr3 was not voltage-dependent. If the inhibition of Kv1.3 and Kv1.6 by CNF-Sr3 was not voltage-dependent, then this should be shown or stated in the Results.

Dear Reviewer

We agree that confirmation of I-V curves for confirming the no voltage-dependency of the binding of CNF-Sr3 on hKv1.3 and hKv1.6 (suggested by our previous findings on Shaker K channel) would have been desirable. Unfortunately, these experiments could not be done within the time given to us for responding to the new comments. Therefore, we did not make any change in the manuscript.

Reviewer 2 Report

The authors improved their manuscript and addressed most of my concerns aside of the main drawback that not all Kv1 subtypes have been evaluated.

The size of the toxin makes it indeed very promising and gives it advantages over others. I understand that this is a screen to spark further research on the toxin such as elucidating the mechanism of action, binding determinants etc. If human clones are not available, evaluating the toxin on the Kv1.1 and Kv1.2 channel from for example mice or rat will already provide evidence that the toxin can be used to study selectively Kv1.6 channels. From the number of Kv subtypes evaluated now, the authors can indeed conclude that a subfamily difference is observed, which may be used as a guidance to continue on the development of synthetic variants. The conclusion that the toxin can be used to study Kv1.6 specifically can still not be drawn.

Comments on newly shown data in the revision

In their revised version the authors interpret the Hill number to suggest that the toxin binds in a one-to-one and two-to-one ratio. This is not correct. The Hill number is an indication of cooperativity between binding of molecules. In the case of a Hill number of 1, still two toxins can bind but they do not exert a cooperative effect, meaning the binding of the second toxin is not affected by the binding of the first. In this case the Hill number is one but the presence of 2 binding positions will affect the IC50 value.

In their adapted figure the authors show wash-in of the toxin but in the case of Kv1.3 no steady-state is reached and wash-in should have taken longer. This might contribute to the different Hill number of the concentration-effect curve between Kv1.6 and Kv1.3. Within the time frame shown, there is no wash-out of the toxin, were longer wash-out periods evaluated and can you comment on the absence of current recovery?

In my opinion, the approval of the procedure/work with the oocytes need to be mentioned in the material&methods section of the manuscript.

Author Response

The authors improved their manuscript and addressed most of my concerns aside of the main drawback that not all Kv1 subtypes have been evaluated.

The size of the toxin makes it indeed very promising and gives it advantages over others. I understand that this is a screen to spark further research on the toxin such as elucidating the mechanism of action, binding determinants etc. If human clones are not available, evaluating the toxin on the Kv1.1 and Kv1.2 channel from for example mice or rat will already provide evidence that the toxin can be used to study selectively Kv1.6 channels. From the number of Kv subtypes evaluated now, the authors can indeed conclude that a subfamily difference is observed, which may be used as a guidance to continue on the development of synthetic variants. The conclusion that the toxin can be used to study Kv1.6 specifically can still not be drawn.

Dear Reviewer

We agree that evaluating the toxin on Kv1 and Kv1.2 channels from rat or mouse would give useful information for assessing the probable selectively on Kv1.6. However, we do not have these clones either.

We thank the reviewer for pointing out that the conclusion that the toxin can be used to study Kv1.6 specifically can still not be drawn. We have modified the last sentence of the Abstract in order not to give this interpretation.

Comments on newly shown data in the revision

In their revised version the authors interpret the Hill number to suggest that the toxin binds in a one-to-one and two-to-one ratio. This is not correct. The Hill number is an indication of cooperativity between binding of molecules. In the case of a Hill number of 1, still two toxins can bind but they do not exert a cooperative effect, meaning the binding of the second toxin is not affected by the binding of the first. In this case the Hill number is one but the presence of 2 binding positions will affect the IC50 value.

We appreciate the remark on our interpretation of the difference in Hill numbers. To avoid any misinterpretation we deleted the text “that indicates that CNF-Sr3 inhibits the channels with a toxin:channel stoichiometry of one-to-one and two-to-one, respectively”.

In their adapted figure the authors show wash-in of the toxin but in the case of Kv1.3 no steady-state is reached and wash-in should have taken longer. This might contribute to the different Hill number of the concentration-effect curve between Kv1.6 and Kv1.3. Within the time frame shown, there is no wash-out of the toxin, were longer wash-out periods evaluated and can you comment on the absence of current recovery?

We agree that a longer wash-in period in the case of Kv1.3 might have allowed to reach a clear steady state. We also agree that the uncertainty on the steady state could have contributed to the difference in Hill numbers for the two channels.

No longer wash-out periods were evaluated. The absence of recovery of the currents along the wash-out periods (approximately 10 minutes for both cases) suggests that CNF-Sr3 molecules have a very slow dissociation rate for both channels.

In my opinion, the approval of the procedure/work with the oocytes need to be mentioned in the material&methods section of the manuscript.

We followed your recommendation and a statement on the approval of the procedures that involved frogs was included in section 4.2 of Materials and Methods.

Reviewer 3 Report

The reviewer has no more concerns.

Author Response

There were no comments